# FLGAME: A GAME-THEORETIC DEFENSE AGAINST BACKDOOR ATTACKS IN FEDERATED LEARNING

## ABSTRACT

Federated learning enables the distributed training paradigm, where multiple local clients jointly train a global model without needing to share their local training data. However, recent studies have shown that federated learning provides an additional surface for backdoor attacks. For instance, an attacker can compromise a subset of clients and thus corrupt the global model to incorrectly predict an attacker-chosen target class given any input embedded with the backdoor trigger. Existing defenses for federated learning against backdoor attacks usually detect and exclude the corrupted information from the compromised clients based on a *static* attacker model. Such defenses, however, are less effective when faced with *dynamic* attackers who can strategically adapt their attack strategies. In this work, we model the strategic interaction between the (global) defender and attacker as a minimax game. Based on the analysis of our model, we design an interactive defense mechanism that we call FLGAME. Theoretically, we prove that under mild assumptions, the global model trained with FLGAME under backdoor attacks is close to that trained without attacks. Empirically, we perform extensive evaluations on benchmark datasets and compare FLGAME with multiple state-of-the-art baselines. Our experimental results show that FLGAME can effectively defend against strategic attackers and achieves significantly higher robustness than baselines.

## 1 INTRODUCTION

Federated learning (FL) (McMahan et al., 2017a) aims to train machine learning models (called *global models*) over training data that is distributed across multiple clients (e.g., mobile phones, IoT devices). FL has been widely used in many real-world applications such as finance (Long et al., 2020) and healthcare (Long et al., 2022). FL trains a global model in an iterative manner. In each communication round, a cloud server shares its global model with selected clients; each selected client uses the global model to initialize its local model, then utilizes its local training dataset to train the local model, and finally sends the local model update to the server; the server uses an aggregation rule to aggregate local model updates from clients to update its global model.

Due to the distributed nature of FL, many recent studies (Bhagoji et al., 2019; Bagdasaryan et al., 2020; Baruch et al., 2019; Wang et al., 2020; Kairouz et al., 2021) have shown that it is vulnerable to backdoor attacks. For instance, an attacker can compromise a subset of clients and manipulate their local training datasets to corrupt the global model such that it predicts an attacker-chosen target class for any inputs embedded with a backdoor trigger (Bagdasaryan et al., 2020). To defend against backdoor attacks, many defenses (Sun et al., 2019; Cao et al., 2021a) have been proposed. For example, Sun et al. (2019) proposed to clip the norm of the local model update from each client such that its $L_2$-norm was no larger than a defender-chosen threshold. Cao et al. (2021a) proposed FLTrust in which a server computes a local model update itself and computes its similarity with that of a client as the trust score, which is leveraged when updating the global model. However, all of those defenses consider a *static* attack model where an attacker does not adapt its attack strategies. As a result, they are less effective under adaptive attacks, e.g., Wang et al. (2020) showed that the defenses proposed in (Sun et al., 2019; Blanchard et al., 2017) can be bypassed by appropriately designed attacks.

**Our contribution:** In this work, we propose FLGAME, a game-theoretic defense against backdoor attacks to FL. Specifically, we formulate FLGAME as a minimax game between the server (defender)

and attacker, which enables them to strategically adapt their defense and attack strategies. In the rest of the paper, we use the terms *benign client* to denote a valid/un-compromised client and *genuine score* to quantify the extent to which a client is benign. Our key idea is that the server can compute a genuine score for each client whose value is large (or small) if the client is benign (or compromised) in each communication round. The genuine score serves as a weight for the local model update of the client when used to update the global model. The goal of the defender is to minimize the genuine scores for compromised clients and maximize them for benign ones. To solve the resulting minimax game for the defender, we follow a three-step process consisting of 1) building an auxiliary global model, 2) exploiting it to reverse engineer a backdoor trigger and target class, and 3) inspecting whether the local model of a client will predict an input embedded with the reverse engineered backdoor trigger as the target class to compute a genuine score for the client. Based on the deployed defense, the goal of the attacker is to optimize its attack strategy by maximizing the effectiveness of the backdoor attack. Our key observation is that the attack effectiveness is determined by two factors: genuine score and the local model of the client. We optimize the attack strategy with respect to those two factors to maximize the effectiveness of backdoor attacks against our defense.

We perform both theoretical analysis and empirical evaluations for FLGAME. Theoretically, we prove that the global model trained with our defense under backdoor attacks is close to that trained without attacks (measured by $L_2$-norm of global model parameters difference). Empirically, we evaluate FLGAME on benchmark datasets to demonstrate its effectiveness under state-of-the-art backdoor attacks. Moreover, we compare it with state-of-the-art baselines. Our results indicate that FLGAME outperforms them by a significant margin. Our key contributions can be summarized as follows:

- We propose a game-theoretic defense FLGAME. We formulate FLGAME as a minimax game between the defender and attacker, which enables them to strategically optimize their defense and attack strategies.

- We theoretically analyze the robustness of FLGAME. In particular, we show that the global model trained with FLGAME under backdoor attacks is close to that without attacks.

- We perform a systematic evaluation of FLGAME on benchmark datasets and demonstrate that FLGAME significantly outperforms state-of-the-art baselines.

## 2 RELATED WORK

**Backdoor attacks on federated learning:** In backdoor attacks to FL (Bhagoji et al., 2019; Bagdasaryan et al., 2020; Baruch et al., 2019; Wang et al., 2020; Zhang et al., 2022b), an attacker aims to make a global model predict a target class for any input embedded with a backdoor trigger via compromised clients. For instance, Bagdasaryan et al. (2020) proposed scaling attack in which an attacker uses a mix of backdoored and clean training examples to train its local model and then scales the local model update by a factor before sending it to the server. Xie et al. (2019) proposed distributed backdoor attack to FL. Roughly speaking, the idea is to decompose a backdoor trigger into different sub-triggers and embed each of them to the local training dataset of different compromised clients. In our work, we will leverage those attacks to perform strategic backdoor attacks to our defense.

**Defenses for Federated learning against backdoor attacks:** Many defenses (Sun et al., 2019; Cao et al., 2021a; Ozdayi et al., 2021; Wu et al., 2020; Rieger et al., 2022; Nguyen et al., 2022) were proposed to mitigate backdoor attacks to FL. For instance, Sun et al. (2019) proposed norm-clipping which clips the norm of the local model update of a client such that its norm is no larger than a threshold. They also extended differential privacy (Dwork et al., 2006; Abadi et al., 2016; McMahan et al., 2017b) to mitigate backdoor attacks to federated learning. The idea is to clip the local model update and add Gaussian noise to it. Cao et al. (2021a) proposed FLTrust which leveraged the similarity of the local model update of a client with that computed by the server itself on its clean dataset. Other defenses include Byzantine-robust FL methods such as Krum (Blanchard et al., 2017), Trimmed Mean (Yin et al., 2018), and Median (Yin et al., 2018). However, all of those defenses consider a static attacker model. As a result, they become less effective against dynamic attackers who strategically adapt their attack strategies.

Another line of research focuses on detecting malicious clients (Li et al., 2020a; Zhang et al., 2022a). For instance, Li et al. (2020a) proposed to train a variational autoencoder (VAE) and use its reconstruction loss on the local model update of a client to detect malicious clients. However, those defenses need to collect many local model updates from a client to make confident detection. As a result, the global model may already be backdoored before those clients are detected. Two recent studies (Cao et al., 2021b; Xie et al., 2021) proposed certified defenses against compromised clients. However, they can only tolerate a moderate fraction of malicious clients (e.g., less than 10%) as shown in their experimental results.

## 3 BACKGROUND ON FEDERATED LEARNING AND THREAT MODEL

### 3.1 FEDERATED LEARNING

Suppose $\mathcal{S}$ is a set of clients. We use $\mathcal{D}_i$ to denote the local training dataset of the client $i \in \mathcal{S}$. In the $t$th communication round, the server first sends the current global model (denoted as $\Theta^t$) to each client. Then, each client $i$ trains a local model (denoted as $\Theta_i^t$) by finetuning the global model $\Theta^t$ using its local training dataset $\mathcal{D}_i$. For simplicity, we use $\mathbf{z} = (\mathbf{x}, y)$ to denote a training example in $\mathcal{D}_i$, where $\mathbf{x}$ is the training input (e.g., an image) and $y$ is its ground truth label. Given $\mathcal{D}_i$ and the global model $\Theta^t$, we denote a loss function $\mathcal{L}(\mathcal{D}_i; \Theta^t) = \frac{1}{|\mathcal{D}_i|} \sum_{\mathbf{z} \in \mathcal{D}_i} \ell(\mathbf{z}; \Theta^t)$ where $\ell$ is a loss function (e.g., cross-entropy loss). The client can use gradient descent to update its local model based on the global model and its local training dataset, i.e., $\Theta_i^t = \Theta^t - \eta_l \frac{\partial \mathcal{L}(\mathcal{D}_i; \Theta^t)}{\partial \Theta^t}$, where $\eta_l$ is the learning rate of the local model. Note that the client can also use stochastic gradient descent to update its local model on its local training dataset as illustrated in our experiments. Then, the client sends $g_i^t = \Theta_i^t - \Theta_i$ (called *local model update*) to the server. Note that it is equivalent for the client to send a local model or local model update to the server as $\Theta_i^t = \Theta^t + g_i^t$. After receiving the local model updates from all clients, the server can aggregate them based on an aggregation rule $\mathcal{R}$ (e.g., FedAvg) to update its glocal model, i.e., we have:

$$\Theta^{t+1} = \Theta^t + \eta \mathcal{R}(g_1^t, g_2^t, \cdots, g_{|\mathcal{S}|}^t), \qquad (1)$$

where $|\mathcal{S}|$ represents the number of clients and $\eta$ is the learning rate of the global model.

### 3.2 THREAT MODEL

We consider the backdoor attack proposed in previous work (Bagdasaryan et al., 2020; Xie et al., 2019). In particular, we assume an attacker can compromise a set of clients (denoted as $\mathcal{S}_a$). To perform the backdoor attack, the attacker first selects a backdoor trigger $\delta$ and a target class $y^{tc}$. For each client $i \in \mathcal{S}_a$ in the $t$th ($t = 1, 2, \cdots$) communication round, the attacker can choose an arbitrary fraction (denoted as $r_i^t$) of training examples from the local training dataset of the client, embed the backdoor trigger $\delta$ to those training inputs, and relabel them as the target class $y^{tc}$. Those backdoored training examples are used to augment the local training dataset of the client. In our game-theoretic framework, we will optimize $r_i^t$ for the compromised client $i$ in each communication round to make the backdoor attack more effective under our defense.

We consider that the server itself has a small clean training dataset (denoted as $\mathcal{D}_s$), which could be collected from the same or different domains of the local training datasets of clients. Moreover, we consider the case that the server does not have any information on each client except their local model updates in each communication round.

## 4 FLGAME: A GAME-THEORETIC DEFENSE AGAINST BACKDOORS

**Overview:** Our idea is to formulate FLGAME as a minimax game between the defender and attacker, solving which enables them to respectively optimize their strategies. In particular, the defender computes a genuine score for each client in each communication round. The goal of the defender is to maximize the genuine score for a benign client and minimize it for a compromised one. Given the genuine score for each client, we use a weighted average over all the local model updates to update the global model, i.e., we have

$$\Theta^{t+1} = \Theta^t + \eta \frac{1}{\sum_{i \in \mathcal{S}} p_i^t} \sum_{i \in \mathcal{S}} p_i^t g_i^t, \qquad (2)$$

where $p_i^t$ is the genuine score for client $i$ in the $t$th communication round and $\eta$ is the learning rate of the global model. The goal of the attacker is to maximize its attack effectiveness, which is determined by two components based on Equation 2: genuine scores and local models of compromised clients. In our framework, the attacker will optimize the tradeoff between those two components to maximize the effectiveness of its backdoor attacks against our defense.

## 4.1 FORMULATING FLGAME AS A MINIMAX GAME

**Computing the genuine score for client** $i$**:** To compute $p_i^t$, our key observation is that the local model of a compromised client is more likely to predict the target class for a trigger-embedded input compared with that of a benign client. However, the key challenge is that the server does not know the backdoor trigger and target class adopted by the attacker. To overcome the challenge, the server can reverse engineer a backdoor trigger $\delta_{re}$ and target class $y_{re}^{tc}$ (we will discuss more details in the next subsection). Recall that the client $i$ sends its local model update $g_i^t$ to the server, the local model of the client $i$ can be computed as $\Theta_i^t = \Theta^t + g_i^t$. Then, we can compute $p_i^t$ as follows:

$$p_i^t = 1 - \frac{1}{|\mathcal{D}_s|} \sum_{\mathbf{x} \in \mathcal{D}_s} \mathbb{I}(G(\mathbf{x} \oplus \delta_{re}; \Theta_i^t) = y_{re}^{tc}), \tag{3}$$

where $\mathbb{I}$ is an indicator function, $\mathcal{D}_s$ is the clean training dataset of the server, $\mathbf{x} \oplus \delta_{re}$ is a trigger-embedded input, and $G(\mathbf{x} \oplus \delta_{re}; \Theta_i^t)$ represents the predicted label of the local model $\Theta_i^t$ for $\mathbf{x} \oplus \delta_{re}$. Roughly speaking, the genuine score for a client is small if its local model predicts a large fraction of inputs embedded with the reverse engineered backdoor trigger as the target class.

**The optimization problem for the defender:** The server aims to reverse engineer the backdoor trigger $\delta_{re}$ and target class $y_{re}^{tc}$ such that the genuine scores for compromised clients are minimized while those for benign clients are maximized. Formally, we have the following optimization problem:

$$\min_{\delta_{re}, y_{re}^{tc}} \sum_{i \in \mathcal{S}_a} p_i^t - \sum_{j \in \mathcal{S} \setminus \mathcal{S}_a} p_j^t. \tag{4}$$

**The optimization problem for the attacker:** The goal of an attacker is to maximize its attack effectiveness. Based on Equation 2, the attacker needs to: 1) maximize the genuine scores for compromised clients while minimizing them for benign ones, i.e., $\max(\sum_{i \in \mathcal{S}_a} p_i^t - \sum_{j \in \mathcal{S} \setminus \mathcal{S}_a} p_j^t)$, and 2) make the local models of compromised clients predict an input embedded with the attacker-chosen backdoor trigger $\delta$ as the target class $y^{tc}$. To perform the backdoor attack in the $t$th communication round, the attacker embeds the backdoor to a certain (denoted as $r_i^t$) fraction of training examples in the local training dataset of the client and uses them to augment it. A larger $r_i^t$ is more likely to make the local model of the client $i$ predict a trigger-embedded input as the target class but also make its genuine score smaller. Therefore, $r_i^t$ measures a tradeoff between them. Formally, the attacker can find the desired tradeoff by solving the following optimization problem:

$$\max_{R^t}(\sum_{i \in \mathcal{S}_a} p_i^t - \sum_{j \in \mathcal{S} \setminus \mathcal{S}_a} p_j^t + \lambda \sum_{i \in \mathcal{S}_a} r_i^t), \tag{5}$$

where $R^t = \{r_i^t | i \in \mathcal{S}_a\}$ and $\lambda$ is a hyperparameter to balance the two terms.

**Minimax game between the defender and the attacker:** Given the optimization problems solved by the defender and attacker, we have the following minimax game:

$$\min_{\delta_{re}, y_{re}^{tc}} \max_{R^t}(\sum_{i \in \mathcal{S}_a} p_i^t - \sum_{j \in \mathcal{S} \setminus \mathcal{S}_a} p_j^t + \lambda \sum_{i \in \mathcal{S}_a} r_i^t). \tag{6}$$

Note that $r_i^t$ ($i \in \mathcal{S}_a$) is chosen by the attacker and thus we can add $r_i^t$ to the objective function in Equation 4 without influencing its solution given the local model updates of clients.

## 4.2 SOLVING THE MINIMAX GAME BY THE DEFENDER

To solve the minimax game in Equation 6 for the defender, our idea is to construct an *auxiliary global model* and then reverse engineer the backdoor trigger and target class based on it.

**Constructing an auxiliary global model:** Suppose $g_i^t$ is the local model update from each client $i \in \mathcal{S}$. Our auxiliary global model is constructed as follows: $\Theta_a^t = \Theta^t + \frac{1}{|\mathcal{S}|} \sum_{i \in \mathcal{S}} g_i^t$. Our intuition is that such aggregated global model is very likely to predict an input embedded with the backdoor trigger $\delta$ as the target class $y^{tc}$ under backdoor attacks.

**Reverse engineering the backdoor trigger and target class:** Given the auxiliary global model, we can use arbitrary methods to reverse engineer the backdoor trigger and target class. Roughly speaking, the goal is to find the backdoor trigger and target class such that the genuine scores for benign clients are large but they are small for compromised clients. For instance, we can leverage Neural Cleanse (Wang et al., 2019), which is the state-of-the-art method to reverse engineer a backdoor trigger and target class. Roughly speaking, Neural Cleanse views each class ($c = 1, 2, \cdots, C$ and $C$ is the total number of classes in the classification task) as a potential target class and finds a perturbation $\delta_c$ with a small $L_1$-norm such that any inputs embedded with it will be classified as the class $c$. We view the trigger with the smallest $L_1$-norm as the backdoor trigger and view the corresponding class as the target class. Formally, we have $y_{re}^{tc} = \arg\min_c \|\delta_c\|_1$ and $\delta_{re} = \delta_{y_{re}^{tc}}$.

The complete algorithm of our FLGAME is shown in Algorithm 1 in Appendix.

### 4.3 SOLVING THE MINIMAX GAME BY THE ATTACKER

The goal of the attacker is to find $r_i^t$ for each client $i \in \mathcal{S}_a$ such that the loss function in Equation 6 is maximized. As the attacker does not know the genuine scores of benign clients, the attacker can find $r_i^t$ to maximize $p_i^t + \lambda r_i^t$ for client $i \in \mathcal{S}_a$ to approximately solve the optimization problem in Equation 6. However, the key challenge is that the attacker does not know the reverse engineered backdoor trigger $\delta_{re}$ and the target class $y_{re}^{tc}$ of the defender to compute the genuine score for client $i$. In response, the attacker can use the backdoor trigger $\delta$ and target class $y^{tc}$ chosen by itself. Moreover, the attacker reserves a certain fraction (e.g., 10%) of training data from its local training dataset $\mathcal{D}_i$ as the validation dataset (denoted as $\mathcal{D}_i^{rev}$) to find the best $r_i^t$.

**Estimating a genuine score for a given $r_i^t$:** For a given $r_i^t$, the client $i$ can embed the backdoor to $r_i^t$ fraction of training examples in $\mathcal{D}_i \setminus \mathcal{D}_i^{rev}$ and then use those backdoored training examples to augment $\mathcal{D}_i \setminus \mathcal{D}_i^{rev}$ to train a local model (denoted as $\tilde{\Theta}_i^t$). Then, the genuine score can be estimated as $\tilde{p}_i^t = 1 - \frac{1}{|\mathcal{D}_i^{rev}|} \sum_{\mathbf{x} \in \mathcal{D}_i^{rev}} \mathbb{I}(G(\mathbf{x} \oplus \delta; \tilde{\Theta}_i^t) = y^{tc})$, where $G(\mathbf{x} \oplus \delta; \tilde{\Theta}_i^t)$ is the predicted label by the global model $\tilde{\Theta}_i^t$ for the trigger-embedded input $\mathbf{x} \oplus \delta$.

**Searching an optimal $r_i^t$:** The client can use grid search to find $r_i^t$ that achieves the largest $\tilde{p}_i^t + \lambda r_i^t$. After estimating the optimal $r_i^t$, client $i$ can embed the backdoor to $r_i^t$ fraction of training examples to augment the local training dataset, train a local model, and send the local model update to the server.

The complete algorithm for each compromised client is shown in Algorithm 2 in Appendix.

## 5 THEORETICAL ANALYSIS OF FLGAME

This section provides a theoretical analysis of FLGAME under backdoor attacks. In particular, we derive an upper bound for the $L_2$-norm of the difference between the parameters of the global models with and without attacks. To analyze the robustness of FLGAME, we make the following assumptions on the loss function used by the clients, which are commonly used in the analysis of previous studies (Li et al., 2020b; Wang & Joshi, 2021; Fallah et al., 2020; Reisizadeh et al., 2020) on federated learning.

**Assumption 1.** *The loss function is $\mu$-strongly convex with $L$-Lipschitz continuous gradient. Formally, we have the following for arbitrary $\Theta$ and $\Theta'$:*

$$(\nabla_\Theta \ell(\mathbf{z}; \Theta) - \nabla_{\Theta'} \ell(\mathbf{z}; \Theta'))^T (\Theta - \Theta') \geq \mu \|\Theta - \Theta'\|_2^2, \tag{7}$$

$$\|\nabla_\Theta \ell(\mathbf{z}; \Theta) - \nabla_{\Theta'} \ell(\mathbf{z}; \Theta')\|_2 \leq L \|\Theta - \Theta'\|_2, \tag{8}$$

*where $\mathbf{z}$ is an arbitrary training example.*

**Assumption 2.** *We assume the gradient $\nabla_\Theta \ell(\mathbf{z}; \Theta)$ is bounded with respect to $L_2$-norm for arbitrary $\Theta$ and $\mathbf{z}$, i.e., there exists some $M \geq 0$ such that*

$$\|\nabla_\Theta \ell(\mathbf{z}; \Theta)\|_2 \leq M. \tag{9}$$

Suppose $\Theta_c^t$ is the global model trained by FLGAME without any attacks in the $t$th communication round, i.e., each client $i \in \mathcal{S}$ uses its clean local training dataset $\mathcal{D}_i$ to train a local model. Moreover, we assume gradient descent with a local model learning rate 1 is used by each client to train its local model. Suppose $q_i^t$ is the genuine score for client $i$ without attacks. Moreover, we denote $\beta_i^t = \frac{q_i^t}{\sum_{i \in \mathcal{S}} q_i^t}$ as the normalized genuine score for client $i$. To perform the backdoor attack, we assume a compromised client $i$ can embed the backdoor trigger to $r_i^t$ fraction of training examples in the local training dataset of the client and relabel them as the target class. Those backdoored training examples are used to augment the local training dataset of the client. Suppose $\Theta^t$ is the global model under the backdoor attack in the $t$th communication round with our defense. We denote $\alpha_i^t = \frac{p_i^t}{\sum_{i \in \mathcal{S}} p_i^t}$ as the normalized genuine score for client $i$ with attacks in the $t$th communication round. We prove the following robustness guarantee for FLGAME:

**Lemma 1** (Robustness Guarantee for One Communication Round). *Suppose Assumptions 1 and 2 hold. Moreover, we assume $(1 - r^t)\beta_i^t \leq \alpha_i^t \leq (1 + r^t)\beta_i^t$, where $i \in \mathcal{S}$ and $r^t = \sum_{j \in \mathcal{S}_a} r_j^t$. Then, we have:*

$$\left\| \Theta^{t+1} - \Theta_c^{t+1} \right\|_2 \leq \sqrt{1 - \eta\mu + 2\eta\gamma^t + \eta^2 L^2 + 2\eta^2 L\gamma^t} \left\| \Theta^t - \Theta_c^t \right\|_2$$
$$+ \sqrt{2\eta\gamma^t(1 + \eta L + 2\eta\gamma^t)} + 2\eta r^t M, \tag{10}$$

*where $\eta$ is the learning rate of the global model, $L$ and $\mu$ are defined in Assumption 1, $\gamma^t = \sum_{i \in \mathcal{S}_a} \alpha_i^t r_i^t M$, and $M$ is defined in Assumption 2.*

*Proof sketch.* Our idea is to decompose $\left\| \Theta^{t+1} - \Theta_c^{t+1} \right\|_2$ into two terms. Then, we derive an upper bound for each term based on the change of the local model updates of clients under backdoor attacks and the properties of the loss function. As a result, our derived upper bound relies on $r_i^t$ for each client $i \in \mathcal{S}_a$, parameters $\mu$, $L$, and $M$ in our assumptions, as well as the parameter difference of the global models in the previous iteration, i.e., $\|\Theta^t - \Theta_c^t\|_2$. Our complete proof can be found in Appendix A.1. $\square$

In the above lemma, we derive an upper bound of $\left\| \Theta^{t+1} - \Theta_c^{t+1} \right\|_2$ with respect to $\left\| \Theta^t - \Theta_c^t \right\|_2$ for one communication round. In the next theorem, we derive an upper bound of $\|\Theta^t - \Theta_c^t\|_2$ as $t \to \infty$. We iterative apply Lemma 1 for successive values of $t$ and have the following theorem:

**Theorem 1** (Robustness Guarantee). *Suppose Assumptions 1 and 2 hold. Moreover, we assume $(1 - r^t)\beta_i^t \leq \alpha_i^t \leq (1 + r^t)\beta_i^t$ for $i \in \mathcal{S}$, $\gamma^t \leq \gamma$ and $r^t \leq r$ hold for all communication round $t$, and $\mu > 2\gamma$, where $r^t = \sum_{j \in \mathcal{S}_a} r_j^t$ and $\gamma^t = \sum_{i \in \mathcal{S}_a} \alpha_i^t r_i^t M$. Let the global model learning rate by chosen as $0 < \eta < \frac{\mu - 2\gamma}{L^2 + 2L\gamma}$. Then, we have:*

$$\left\| \Theta^t - \Theta_c^t \right\|_2 \leq \frac{\sqrt{2\eta\gamma(1 + \eta L + 2\eta\gamma)} + 2\eta r M}{1 - \sqrt{1 - \eta\mu + 2\eta\gamma + \eta^2 L^2 + 2\eta^2 L\gamma}} \tag{11}$$

*holds as $t \to \infty$.*

*Proof sketch.* Given the conditions that $\gamma^t \leq \gamma$ and $r^t \leq r$ as well as the fact that the right-hand side of Equation 10 is monotonic with respect to $\gamma^t$ and $r^t$, we can replace $\gamma^t$ and $r^t$ in Equation 10 with $\gamma$ and $r$. Then, we iterative apply the equation for successive values of $t$. When $0 < \eta < \frac{\mu - 2\gamma}{L^2 + 2L\gamma}$, we have $0 < 1 - \eta\mu + 2\eta\gamma + \eta^2 L^2 + 2\eta^2 L\gamma < 1$. By letting $r \to \infty$, we can reach the conclusion. The complete proof can be found in Appendix A.2. $\square$

When $r_i^t = 0$ for $\forall i, \forall t$, we have $\gamma = 0$ and $r = 0$. Thus, the upper bound in Equation 11 becomes 0.

## 6 EXPERIMENTS

### 6.1 EXPERIMENTAL SETUP

**Datasets and global models:** We use two datasets: MNIST (LeCun et al., 2010) and CIFAR10 (Krizhevsky, 2009) for FL tasks. MNIST has 60,000 training and 10,000 testing images, each of

which has a size of $28 \times 28$ belonging to one of 10 classes. CIFAR10 consists of 50,000 training and 10,000 testing images with a size of $32 \times 32$. Each image is categorized into one of 10 classes. For each dataset, we randomly sample 90% of training data for clients, and the remaining 10% of training data is reserved to evaluate our defense when the clean training dataset of the server is from the same domain as those of clients. We use a CNN with two convolution layers (detailed architecture can be found in Table 5 in Appendix) and ResNet-18 (He et al., 2016) which is pre-trained on ImageNet (Deng et al., 2009) as the global models for MNIST and CIFAR10.

**FL settings:** We consider two settings: local training datasets of clients are independently and identically distributed (i.e., IID), and not IID (i.e., non-IID). In IID setting, we randomly distribute the training data to each client. In non-IID, we follow the previous work (Fang et al., 2020) to distribute training data to clients. In particular, they use a parameter $q$ to control the degree of non-IID, which models the probability that training images from a class are distributed to a particular client (or a set of clients). We set $q = 0.5$ by following (Fang et al., 2020). Unless otherwise mentioned, we consider the IID setting. Moreover, we train a global model based on 10 clients for 200 iterations with a global model learning rate $\eta = 1.0$. In each communication round, we use SGD to train the local model of each client for two epochs with a local model learning rate 0.01. Moreover, we consider all clients are selected in each communication round.

**Backdoor attack settings:** We consider state-of-the-art backdoor attacks to federated learning, i.e., Scaling attack Bagdasaryan et al. (2020) and DBA (Xie et al., 2019). We use the same backdoor trigger and target class as used in those works. By default, we assume 60% of clients are compromised by an attacker. We set the scaling parameter to be #total clients/($\eta \times$#compromised clients) following Bagdasaryan et al. (2020). When the attacker solves the minimax game in Equation 6, we set the default $\lambda = 1$. We will explore its impact in our experiments. We randomly sample 10% of the local training data of each compromised client as validation data to search for an optimal $r_i^t$. Moreover, we set the granularity of grid search to be 0.1 when searching for $r_i^t$.

**Baselines:** We compare our defense with the following methods: FedAvg (McMahan et al., 2017a), Krum (Blanchard et al., 2017), Median (Yin et al., 2018), Norm-Clipping (Sun et al., 2019), Differential Privacy (DP) (Sun et al., 2019), and FLTrust (Cao et al., 2021a). FedAvg is non-robust while Krum and Median are two Byzantine-robust baselines. Norm-Clipping clips the $L_2$-norm of local model updates to a given threshold $\mathcal{T}_N$. We set $\mathcal{T}_N = 0.01$ for MNIST and $\mathcal{T}_N = 0.1$ for CIFAR10. DP first clips the $L_2$-norm of a local model update to a threshold $\mathcal{T}_D$ and then adds Gaussian noise. We set $\mathcal{T}_D = 0.05$ for MNIST and $\mathcal{T}_D = 0.5$ for CIFAR10. We set the standard deviation of noise to be 0.01 for both datasets. In FLTrust, the server uses its clean dataset to compute a server model update and assigns a trust score to each client by leveraging the similarity between the server model update and the local model update. We set the clean training dataset of the server to be the same as FLGAME in our comparison. Note that FLTrust is not applicable when the clean training dataset of the server is from a different domain from those of clients.

**Evaluation metrics:** We use *testing accuracy (TA)* and *attack success rate (ASR)* as evaluation metrics. TA is the fraction of clean testing inputs that are correctly predicted. ASR is the fraction of backdoored testing inputs that are predicted as the target class.

**Defense setting:** We consider two settings: in-domain and out-of-domain. For the in-domain setting, we consider the clean training dataset of the server is from the same domain as the local training datasets of clients. We use the reserved data as the clean training dataset of the server for each dataset. For the out-of-domain setting, we consider the server has a clean training dataset that is from the different domains of FL tasks. In particular, we randomly sample 6,000 images from FashionMNIST (Xiao et al., 2017) for MNIST and sample 5,000 images from GTSRB (Houben et al., 2013) for CIFAR10 as the clean training dataset of the server. We adopt Neural Cleanse (Wang et al., 2019) to reverse engineer the backdoor trigger and target class.

## 6.2 EXPERIMENTAL RESULTS

**Our FLGAME consistently outperforms existing defenses:** Table 1 and Table 2 show the results of FLGAME compared with existing defenses under IID and non-IID settings. We have the following observations from the experimental results. First, FLGAME outperforms all existing defenses in terms of ASR. In particular, FLGAME can reduce ASR to random guessing (i.e., ASR of FedAvg under no attacks) in both IID and non-IID settings for clients as well as both in-domain and out-of-

Table 1: **Comparison of FLGAME with existing defenses under Scaling attack. The total number of clients is 10 with 60% compromised. The best results for defense are bold.**

| Datasets | Metrics | FedAvg (No attacks) | Defenses (Under attacks) | | | | | | FLGAME | |
|---|---|---|---|---|---|---|---|---|---|---|
| | | | FedAvg | Krum | Median | Norm-Clipping | DP | FLTrust | In-domain | Out-of-domain |
| MNIST | TA (%) | 99.04 | 98.77 | 98.78 | 99.17 | 95.48 | 92.97 | 97.93 | 98.53 | 98.56 |
| | ASR (%) | 9.69 | 99.99 | 99.99 | 99.97 | 98.54 | 99.45 | 16.01 | 9.72 | **9.68** |
| CIFAR10 | TA (%) | 81.08 | 80.51 | 76.44 | 80.17 | 80.38 | 43.22 | 75.71 | 74.81 | 67.42 |
| | ASR (%) | 8.39 | 99.8 | 99.94 | 99.82 | 99.87 | 99.58 | 99.46 | **8.92** | 9.57 |

Table 2: **Comparison of FLGAME with existing defenses under Scaling attack. The total number of clients is 10 with 60% compromised. The local training datasets of clients are non-IID. The best results for defense are bold.**

| Datasets | Metrics | FedAvg (No attacks) | Defenses (Under attacks) | | | | | | FLGAME | |
|---|---|---|---|---|---|---|---|---|---|---|
| | | | FedAvg | Krum | Median | Norm-Clipping | DP | FLTrust | In-domain | Out-of-domain |
| MNIST | TA (%) | 98.98 | 99.15 | 96.88 | 99.12 | 94.54 | 91.52 | 97.68 | 98.28 | 98.34 |
| | ASR (%) | 9.73 | 99.99 | 85.03 | 99.98 | 98.16 | 99.54 | 19.61 | **10.42** | 10.88 |
| CIFAR10 | TA (%) | 80.25 | 75.35 | 67.66 | 79.54 | 70.18 | 50.79 | 75.08 | 73.43 | 72.37 |
| | ASR (%) | 9.67 | 99.92 | 99.92 | 99.99 | 99.63 | 95.01 | 99.82 | 11.51 | **10.62** |

Table 3: **Comparison of FLGAME with existing defenses under Scaling attack. The total number of clients is 30 with 60% compromised. The best results for defense are bold.**

| Datasets | Metrics | FedAvg (No attacks) | Defenses (Under attacks) | | | | | | FLGAME | |
|---|---|---|---|---|---|---|---|---|---|---|
| | | | FedAvg | Krum | Median | Norm-Clipping | DP | FLTrust | In-domain | Out-of-domain |
| MNIST | TA (%) | 99.02 | 99.09 | 98.16 | 99.01 | 92.77 | 89.77 | 95.27 | 97.81 | 97.64 |
| | ASR (%) | 9.74 | 99.98 | 99.98 | 99.98 | 98.2 | 98.83 | 11.04 | **9.95** | **9.95** |
| CIFAR10 | TA (%) | 80.08 | 79.73 | 72.23 | 79.58 | 79.2 | 50.86 | 67.84 | 73.29 | 60.1 |
| | ASR (%) | 9.14 | 99.82 | 99.97 | 99.85 | 99.87 | 96.53 | 99.28 | **10.44** | 12.87 |

Table 4: **Comparison of FLGAME with existing defenses under DBA attack. The total number of clients is 10 with 60% compromised. The best results for defense are bold.**

| Datasets | Metrics | FedAvg (No attacks) | Defenses (Under attacks) | | | | | | FLGAME | |
|---|---|---|---|---|---|---|---|---|---|---|
| | | | FedAvg | Krum | Median | Norm-Clipping | DP | FLTrust | In-domain | Out-of-domain |
| MNIST | TA (%) | 99.02 | 99.03 | 98.87 | 98.98 | 98.99 | 98.99 | 97.98 | 97.84 | 98.05 |
| | ASR (%) | 9.74 | 100 | 10.06 | 99.81 | 99.75 | 99.73 | 10.02 | **9.56** | 9.68 |
| CIFAR10 | TA (%) | 81.08 | 80.9 | 76.09 | 80.0 | 80.21 | 41.36 | 75.17 | 73.18 | 72.93 |
| | ASR (%) | 8.39 | 93.44 | 94.97 | 91.60 | 91.90 | 86.96 | 66.58 | **8.81** | 9.00 |

domain settings for the server. Intrinsically, FLGAME performs better because our game-theoretic defense enables the defender to optimize its strategy against dynamic, adaptive attacks. We note that FLTrust outperforms other defenses (except FLGAME) in most cases since it exploits a clean training dataset from the same domain as local training datasets of clients. However, FLTrust is not applicable when the server only holds an out-of-domain clean training dataset, while FLGAME can relax such an assumption and will still be applicable. Moreover, our experimental results indicate that FLGAME achieves comparable performance even if the server holds an out-of-domain clean training dataset. In Appendix C.2, we visualize the average genuine (or trust) scores computed by FLGAME (or FLTrust) for compromised and benign clients to further explain why our FLGAME outperforms FLTrust. Second, our FLGAME achieves comparable TA with existing defenses, indicating that our FLGAME preserves the utility of global models.

Table 3 shows the comparison results of FLGAME with existing defenses when the total number of clients is 30. Table 4 shows the comparison results of FLGAME with existing defenses under DBA attack. Our observations are similar, which indicates that FLGAME consistently outperforms existing defenses under different numbers of clients and backdoor attacks.

**Impact of $\lambda$:** $\lambda$ is a hyperparameter used by an attacker when searching for the optimal $r_i^t$ for each compromised client $i$ in each communication round $t$. Figure 1 shows the impact of $\lambda$ on ASR of our FLGAME. The results show that our FLGAME is insensitive to different $\lambda$'s. The reason is that the genuine score for a compromised client is small when $\lambda$ is large, and the local model of a compromised client is less likely to predict a trigger-embedded input as the target class when $\lambda$ is small. As a result, backdoor attacks with different $\lambda$ are ineffective under our FLGAME.

**Impact of the fraction of compromised clients:** Figure 2 shows the impact of the fraction of compromised clients on ASR of our FLGAME and FLTrust. As the results show, our FLGAME is effective for a different fraction of compromised clients in both in-domain and out-of-domain settings. In contrast, FLTrust is ineffective when the fraction of compromised clients is large. For instance, our FLGAME can achieve 9.84% (in-domain) and 10.12% (out-of-domain) ASR even if 80% of clients are compromised on MNIST. Under the same setting, the ASR of FLTrust is 99.95%, indicating that the defense fails.

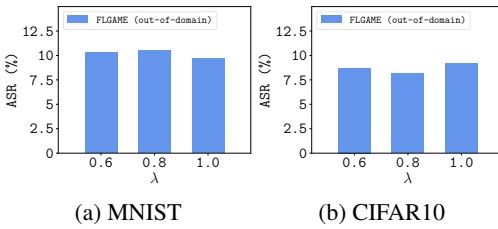

(a) MNIST     (b) CIFAR10

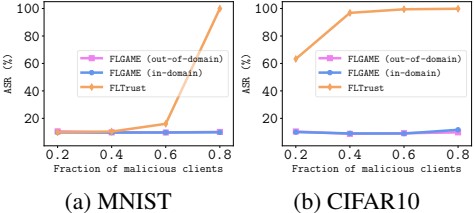

(a) MNIST     (b) CIFAR10

**Figure 1: Impact of $\lambda$ on ASR of FLGAME under Scaling attack. FLGAME is insensitive to various choices of $\lambda$.**

**Figure 2: Impact of the fraction of compromised clients on ASR of FLGAME and FLTrust under Scaling attack.**

## 7   CONCLUSION AND FUTURE WORK

In this work, we propose FLGAME, a general game-theoretic defense against adaptive backdoor attacks to federated learning. Our formulated minimax game enables the defender and attacker to dynamically optimize their strategies. Moreover, we respectively design solutions for both of them to solve the minimax game. Theoretically, we show that the parameters of the global model with the backdoor attack under our FLGAME is close to that without attacks. Empirically, we perform systematic evaluations on benchmark datasets and compare FLGAME with multiple state-of-the-art baselines. Our results demonstrate the effectiveness of FLGAME under strategic backdoor attacks. Moreover, FLGAME achieves significantly higher robustness than baselines.

Interesting future work includes: 1) extending our FLGAME to defend against other attacks to federated learning, and 2) improving FLGAME by designing new methods to reverse engineer the backdoor trigger and target class via exploiting the historical local model updates sent by each client.

ETHICS STATEMENT

We propose a game-theoretic defense against backdoor attacks to federated learning in this work. One potentially harmful effect is that an attacker may leverage our defense to enhance its attack. However, our defense already considers strategic attacks. Therefore, we do not see any explicit ethical issues with our work.

REPRODUCIBILITY STATEMENT

We discuss the reproducibility of our work from two aspects: theoretic analysis and empirical results. For theoretic analysis, we explicitly explain the assumptions that we make in Section 5. We also include the complete proofs for our lemmas and theorems in Appendix. For empirical results, we discuss the details of our experimental setup in Section 6.1, including datasets and global models, federated learning settings, backdoor attack settings, baselines, and their parameter settings, as well as our FLGAME settings. The datasets used in this work are all publicly available. We also add the link to the publicly available codes used in our experiments. We will release our code upon paper acceptance.

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

APPENDIX

# A  COMPLETE PROOFS

## A.1  PROOF OF LEMMA 1

We first present some preliminary lemmas that will be invoked for proving Lemma 1.

**Lemma 2.** *Suppose $\mathcal{D}_i$ is the clean local training dataset of the client $i$. An attacker can inject the backdoor trigger to $r_i^t$ fraction of training examples in $\mathcal{D}_i$ and relabel them as the target class. We use $\mathcal{D}_i'$ to denote the set of backdoored training examples where $r_i^t = \frac{|\mathcal{D}_i'|}{|\mathcal{D}_i|}$. Given two arbitrary $\Theta$ and $\Theta_c$, we let $g_i = \frac{1}{|\mathcal{D}_i \cup \mathcal{D}_i'|} \nabla_\Theta \sum_{\mathbf{z} \in \mathcal{D}_i \cup \mathcal{D}_i'} \ell(\mathbf{z}; \Theta)$ and $h_i = \frac{1}{|\mathcal{D}_i|} \nabla_{\Theta_c} \sum_{\mathbf{z} \in \mathcal{D}_i} \ell(\mathbf{z}; \Theta_c)$. We then have that*

$$(\Theta - \Theta_c)^T (g_i - h_i) \geq (0.5\mu - r_i^t M) \|\Theta - \Theta_c\|_2^2 - r_i^t M, \tag{12}$$

$$\|g_i - h_i\|_2 \leq L \|\Theta - \Theta_c\|_2 + 2r_i^t M. \tag{13}$$

*Proof.* We first prove Equation 12. We have the following relations:

$$(\Theta - \Theta_c)^T (g_i - h_i)$$

$$= (\Theta - \Theta_c)^T \left( \frac{1}{|\mathcal{D}_i \cup \mathcal{D}_i'|} \sum_{\mathbf{z}' \in \mathcal{D}_i \cup \mathcal{D}_i'} \nabla_\Theta \ell(\mathbf{z}'; \Theta) - \frac{1}{|\mathcal{D}_i|} \sum_{\mathbf{z} \in \mathcal{D}_i} \nabla_{\Theta_c} \ell(\mathbf{z}; \Theta_c) \right) \quad \triangleright \text{ definition of } g_i \text{ and } h_i \tag{14}$$

$$= (\Theta - \Theta_c)^T \left( \frac{1}{(1 + r_i^t)|\mathcal{D}_i|} \sum_{\mathbf{z}' \in \mathcal{D}_i \cup \mathcal{D}_i'} \nabla_\Theta \ell(\mathbf{z}'; \Theta) - \frac{1}{|\mathcal{D}_i|} \sum_{\mathbf{z} \in \mathcal{D}_i} \nabla_{\Theta_c} \ell(\mathbf{z}; \Theta_c) \right) \quad \triangleright r_i^t = \frac{|\mathcal{D}_i'|}{|\mathcal{D}_i|} \tag{15}$$

$$= \frac{1}{|\mathcal{D}_i|(1 + r_i^t)} (\Theta - \Theta_c)^T \left( \sum_{\mathbf{z}' \in \mathcal{D}_i \cup \mathcal{D}_i'} \nabla_\Theta \ell(\mathbf{z}'; \Theta) - (1 + r_i^t) \sum_{\mathbf{z} \in \mathcal{D}_i} \nabla_{\Theta_c} \ell(\mathbf{z}; \Theta_c) \right) \tag{16}$$

$$= \frac{1}{|\mathcal{D}_i|(1 + r_i^t)} (\Theta - \Theta_c)^T \left( \sum_{\mathbf{z}' \in \mathcal{D}_i} \nabla_\Theta \ell(\mathbf{z}'; \Theta) - \sum_{\mathbf{z} \in \mathcal{D}_i} \nabla_{\Theta_c} \ell(\mathbf{z}; \Theta_c) \right.$$
$$\left. + \sum_{\mathbf{z}' \in \mathcal{D}_i'} \nabla_\Theta \ell(\mathbf{z}'; \Theta) - r_i^t \sum_{\mathbf{z} \in \mathcal{D}_i} \nabla_{\Theta_c} \ell(\mathbf{z}; \Theta_c) \right) \tag{17}$$

$$= \frac{1}{|\mathcal{D}_i|(1 + r_i^t)} \left( \sum_{\mathbf{z} \in \mathcal{D}_i} (\Theta - \Theta_c)^T (\nabla_\Theta \ell(\mathbf{z}; \Theta) - \nabla_{\Theta_c} \ell(\mathbf{z}; \Theta_c)) \right.$$
$$\left. + (\Theta - \Theta_c)^T \left( \sum_{\mathbf{z}' \in \mathcal{D}_i'} \nabla_\Theta \ell(\mathbf{z}'; \Theta) - r_i^t \sum_{\mathbf{z} \in \mathcal{D}_i} \nabla_{\Theta_c} \ell(\mathbf{z}; \Theta_c) \right) \right) \tag{18}$$

$$\geq \frac{1}{|\mathcal{D}_i|(1 + r_i^t)} \left( \sum_{\mathbf{z} \in \mathcal{D}_i} (\Theta - \Theta_c)^T (\nabla_\Theta \ell(\mathbf{z}; \Theta) - \nabla_{\Theta_c} \ell(\mathbf{z}; \Theta_c)) \right.$$
$$\left. - \|(\Theta - \Theta_c)^T \left( \sum_{\mathbf{z}' \in \mathcal{D}_i'} \nabla_\Theta \ell(\mathbf{z}'; \Theta) - r_i^t \sum_{\mathbf{z} \in \mathcal{D}_i} \nabla_{\Theta_c} \ell(\mathbf{z}; \Theta_c)) \|_1 \right) \quad \triangleright \forall x, x \geq -\|x\|_1 \tag{19}$$

$$\geq \frac{1}{|\mathcal{D}_i|(1 + r_i^t)} \left( \sum_{\mathbf{z} \in \mathcal{D}_i} (\Theta - \Theta_c)^T (\nabla_\Theta \ell(\mathbf{z}; \Theta) - \nabla_{\Theta_c} \ell(\mathbf{z}; \Theta_c)) \right.$$
$$\left. - \|\Theta - \Theta_c\|_2 \cdot \| \sum_{\mathbf{z}' \in \mathcal{D}_i'} \nabla_\Theta \ell(\mathbf{z}'; \Theta) - r_i^t \sum_{\mathbf{z} \in \mathcal{D}_i} \nabla_{\Theta_c} \ell(\mathbf{z}; \Theta_c) \|_2 \right) \quad \triangleright \text{ Cauchy–Schwarz inequality}$$

$$\geq \frac{1}{|\mathcal{D}_i|(1 + r_i^t)} \left( \sum_{\mathbf{z} \in \mathcal{D}_i} (\Theta - \Theta_c)^T (\nabla_\Theta \ell(\mathbf{z}; \Theta) - \nabla_{\Theta_c} \ell(\mathbf{z}; \Theta_c)) \right.$$
$$\left. - \|\Theta - \Theta_c\|_2 \cdot \left( \sum_{\mathbf{z}' \in \mathcal{D}_i'} \|\nabla_\Theta \ell(\mathbf{z}'; \Theta)\|_2 + r_i^t \sum_{\mathbf{z} \in \mathcal{D}_i} \|\nabla_{\Theta_c} \ell(\mathbf{z}; \Theta_c)\|_2 \right) \right) \quad \triangleright \text{ triangle inequality}$$

$$\geq \frac{1}{|\mathcal{D}_i|(1+r_i^t)} (\mu|\mathcal{D}_i| \, \|\Theta - \Theta_c\|_2^2 - 2r_i^t|\mathcal{D}_i|M \, \|\Theta - \Theta_c\|_2) \quad \triangleright \text{Assumption 1} \tag{20}$$

$$= \frac{\mu}{1+r_i^t} \|\Theta - \Theta_c\|_2^2 - \frac{1}{1+r_i^t} 2r_i^t M \, \|\Theta - \Theta_c\|_2) \tag{21}$$

$$\geq 0.5\mu \, \|\Theta - \Theta_c\|_2^2 - 2r_i^t M \, \|\Theta - \Theta_c\|_2 \quad \triangleright r_i^t \in [0,1] \tag{22}$$

$$\geq 0.5\mu \, \|\Theta - \Theta_c\|_2^2 - r_i^t M \, \|\Theta - \Theta_c\|_2^2 - r_i^t M) \tag{23}$$

$$= (0.5\mu - r_i^t M) \, \|\Theta - \Theta_c\|_2^2 - r_i^t M, \tag{24}$$

where Equation 23 holds based on the fact that $-2r_i^t M \, \|\Theta - \Theta_c\|_2 \geq -r_i^t M \, \|\Theta - \Theta_c\|_2^2 - r_i^t M$ for $\forall r_i^t \geq 0$ and $\forall M \geq 0$.

In the following, we prove inequality 13. We have that

$$\|g_i - h_i\|_2$$
$$= \frac{1}{|\mathcal{D}_i|(1+r_i^t)} \| \sum_{\mathbf{z}' \in \mathcal{D}_i \cup \mathcal{D}_i'} \nabla_\Theta \ell(\mathbf{z}'; \Theta) - (1+r_i^t) \sum_{\mathbf{z} \in \mathcal{D}_i} \nabla_{\Theta_c} \ell(\mathbf{z}; \Theta_c) \|_2 \quad \triangleright \text{definition of } g_i \text{ and } h_i$$
$$\tag{25}$$

$$= \frac{1}{|\mathcal{D}_i|(1+r_i^t)} \| \sum_{\mathbf{z}' \in \mathcal{D}_i'} \nabla_\Theta \ell(\mathbf{z}'; \Theta) + \sum_{\mathbf{z}' \in \mathcal{D}_i} \nabla_\Theta \ell(\mathbf{z}'; \Theta) - (1+r_i^t) \sum_{\mathbf{z} \in \mathcal{D}_i} \nabla_{\Theta_c} \ell(\mathbf{z}; \Theta_c) \|_2 \tag{26}$$

$$\leq \frac{1}{|\mathcal{D}_i|(1+r_i^t)} \| \sum_{\mathbf{z}' \in \mathcal{D}_i'} \nabla_\Theta \ell(\mathbf{z}'; \Theta) - r_i^t \sum_{\mathbf{z} \in \mathcal{D}_i} \nabla_{\Theta_c} \ell(\mathbf{z}; \Theta_c) \|_2$$
$$+ \frac{1}{|\mathcal{D}_i|(1+r_i^t)} \| \sum_{\mathbf{z}' \in \mathcal{D}_i} \nabla_\Theta \ell(\mathbf{z}'; \Theta) - \sum_{\mathbf{z} \in \mathcal{D}_i} \nabla_{\Theta_c} \ell(\mathbf{z}; \Theta_c) \|_2 \quad \triangleright \text{triangle inequality} \tag{27}$$

$$\leq \frac{1}{1+r_i^t} (2r_i^t M + L\|\Theta - \Theta_c\|_2) \tag{28}$$

$$\leq 2r_i^t M + L\|\Theta - \Theta_c\|_2 \quad \triangleright r_i^t \in [0,1] \tag{29}$$

where Equation 28 is due to Assumption 1 and 2. $\qquad \square$

Given Lemma 2, we prove Lemma 1 as follows. Recall that we have $\alpha_i^t = \frac{p_i^t}{\sum_{i \in S} p_i^t}$ and $\beta_i^t = \frac{q_i^t}{\sum_{i \in S} q_i^t}$.

$$\|\Theta^{t+1} - \Theta_c^{t+1}\|_2 \tag{30}$$

$$= \|\Theta^t - \eta \sum_{i \in \mathcal{S}} \alpha_i^t g_i^t - (\Theta_c^t - \eta \sum_{i \in \mathcal{S}} \beta_i^t h_i^t) \|_2 \quad \triangleright \text{gradient descent for } \Theta^{t+1} \text{ and } \Theta_c^{t+1} \tag{31}$$

$$= \|\Theta^t - \eta \sum_{i \in \mathcal{S}} \alpha_i^t g_i^t - (\Theta_c^t - \eta \sum_{i \in \mathcal{S}} (\alpha_i^t + \beta_i^t - \alpha_i^t) h_i^t) \|_2 \tag{32}$$

$$= \|\Theta^t - \Theta_c^t - \eta \sum_{i \in \mathcal{S}} \alpha_i^t (g_i^t - h_i^t) + (\eta \sum_{i \in \mathcal{S}} (\beta_i^t - \alpha_i^t) h_i^t) \|_2 \quad \triangleright \text{rearranging Equation 32} \tag{33}$$

$$\leq \|\Theta^t - \Theta_c^t - \eta \sum_{i \in \mathcal{S}} \alpha_i^t (g_i^t - h_i^t) \|_2 + \|\eta \sum_{i \in \mathcal{S}} (\beta_i^t - \alpha_i^t) h_i^t \|_2. \quad \triangleright \text{triangle inequality} \tag{34}$$

Next, we respectively derive an upper bound for the first and second terms in Equation 34. To derive the upper bound for the first term, we have that

$$\|\Theta^t - \Theta_c^t - \eta \sum_{i \in \mathcal{S}} \alpha_i^t (g_i^t - h_i^t) \|_2^2$$

$$= \|\Theta^t - \Theta_c^t\|_2^2 - 2\eta(\Theta^t - \Theta_c^t)^T (\sum_{i \in \mathcal{S}} \alpha_i^t (g_i^t - h_i^t)) + \eta^2 \| \sum_{i \in \mathcal{S}} \alpha_i^t (g_i^t - h_i^t) \|_2^2 \tag{35}$$

$$= S_1 + S_2 + S_3, \tag{36}$$

where $S_1 = \|\Theta^t - \Theta_c^t\|_2^2$, $S_2 = -2\eta(\Theta^t - \Theta_c^t)^T(\sum_{i\in\mathcal{S}}\alpha_i^t(g_i^t - h_i^t))$, and $S_3 = \eta^2 \|\sum_{i\in\mathcal{S}}\alpha_i^t(g_i^t - h_i^t)\|_2^2$. Next, we will bound $S_2$ and $S_3$. We denote $\gamma^t = \sum_{i\in\mathcal{S}_a}\alpha_i^t r_i^t M$. Note that we have $\gamma^t = \sum_{i\in\mathcal{S}}\alpha_i^t r_i^t M$ since $r_i^t = 0$ for $\forall i \in \mathcal{S}\setminus\mathcal{S}_a$. We bound $S_2$ as follows.

$$S_2$$

$$= -2\eta(\Theta^t - \Theta_c^t)^T(\sum_{i\in\mathcal{S}}\alpha_i^t(g_i^t - h_i^t)) \tag{37}$$

$$= -2\eta\sum_{i\in\mathcal{S}}\alpha_i^t(\Theta^t - \Theta_c^t)^T(g_i^t - h_i^t) \tag{38}$$

$$\leq -2\eta\sum_{i\in\mathcal{S}}\alpha_i^t((0.5\mu - r_i^t M)\|\Theta^t - \Theta_c^t\|_2^2 - r_i^t M) \tag{39}$$

$$= -2\eta((0.5\mu - \sum_{i\in\mathcal{S}}\alpha_i^t r_i^t M)\|\Theta^t - \Theta_c^t\|_2^2 - \sum_{i\in\mathcal{S}_a}\alpha_i^t r_i^t M) \tag{40}$$

$$= (-\eta\mu + 2\eta\gamma^t)\|\Theta^t - \Theta_c^t\|_2^2 + 2\eta\gamma^t, \quad \triangleright\text{definition of }\gamma^t \tag{41}$$

where inequality 39 holds by Lemma 2 and the fact that $\eta, \alpha_i^t \geq 0$. We bound $S_3$ as follows.

$$S_3$$

$$= \eta^2\|\sum_{i\in\mathcal{S}}\alpha_i^t(g_i^t - h_i^t)\|_2^2 \tag{42}$$

$$\leq \eta^2(\sum_{i\in\mathcal{S}}\alpha_i^t\|(g_i^t - h_i^t)\|_2)^2 \tag{43}$$

$$\leq \eta^2(\sum_{i\in\mathcal{S}}\alpha_i^t(2r_i^t M + L\|\Theta - \Theta_c\|_2)^2 \quad \triangleright \text{Lemma 2} \tag{44}$$

$$= \eta^2(2\gamma^t + L\|\Theta - \Theta_c\|_2)^2 \tag{45}$$

$$= \eta^2(L^2\|\Theta - \Theta_c\|_2^2 + 4\gamma^t L\|\Theta - \Theta_c\|_2 + 4[\gamma^t]^2) \tag{46}$$

$$\leq \eta^2(L^2\|\Theta - \Theta_c\|_2^2 + 2\gamma^t L\|\Theta - \Theta_c\|_2^2 + 2L\gamma^t + 4[\gamma^t]^2) \tag{47}$$

$$= \eta^2 \cdot ((L^2 + 2L\gamma^t)\cdot\|\Theta - \Theta_c\|_2^2 + 2L\gamma^t + 4[\gamma^t]^2) \tag{48}$$

where Equation 47 is based on the fact that $4\gamma^t L\|\Theta - \Theta_c\|_2 \leq 2\gamma^t L\|\Theta - \Theta_c\|_2^2 + 2\gamma^t L$ when $\gamma^t L \geq 0$.

Given the upper bounds of $S_2$ and $S_3$, we can bound $\|\Theta^t - \Theta_c^t - \eta\sum_{i\in\mathcal{S}}\alpha_i^t(g_i^t - h_i^t)\|_2^2$ as follows.

$$\|\Theta^t - \Theta_c^t - \eta\sum_{i\in\mathcal{S}}\alpha_i^t(g_i^t - h_i^t)\|_2^2 \tag{49}$$

$$= S_1 + S_2 + S_3 \tag{50}$$

$$\leq \|\Theta - \Theta_c\|_2^2 + (-\eta\mu + 2\eta\gamma^t)\|\Theta^t - \Theta_c^t\|_2^2 + 2\eta\gamma^t$$
$$+ (\eta^2 L^2 + \eta^2 2L\gamma^t)\|\Theta^t - \Theta_c^t\|_2^2 + \eta^2 2L\gamma^t + \eta^2 4[\gamma^t]^2 \tag{51}$$

$$= (1 - \eta\mu + 2\eta\gamma^t + \eta^2 L^2 + 2\eta^2 L\gamma^t)\|\Theta^t - \Theta_c^t\|_2^2 + 2\eta\gamma^t + 2\eta^2 L\gamma^t + 4\eta^2[\gamma^t]^2 \tag{52}$$

Next, we will derive an upper bound for $\|\eta\sum_{i\in\mathcal{S}}(\beta_i^t - \alpha_i^t)h_i^t\|_2$. We denote $r^t = \sum_{i\in\mathcal{S}_a}r_i^t$. Note that we have that $r^t = \sum_{i\in\mathcal{S}}r_i^t$ also holds since $r_i^t = 0$ for $\forall i \in \mathcal{S}\setminus\mathcal{S}_a$. Given the assumption that $(1 - r^t)\alpha_i^t \leq \beta_i^t \leq (1 + r^t)\alpha_i^t$, we have

$$\|\eta\sum_{i\in\mathcal{S}}(\beta_i^t - \alpha_i^t)h_i^t\|_2 \leq \eta\sum_{i\in\mathcal{S}}|\beta_i^t - \alpha_i^t|\|h_i^t\|_2 \leq 2\eta r^t M, \tag{53}$$

where the first inequality is due to triangle inequality and the second inequality is based on the assumption that $\|h_i^t\|_2 \leq M$. Therefore, we have:

$$\|\Theta^{(t+1)} - \Theta_c^{(t+1)}\|_2$$

$$\leq \|\Theta^t - \Theta_c^t - \eta \sum_{i \in \mathcal{S}} \alpha_i^t(g_i^t - h_i^t)\|_2^2 + \|\eta \sum_{i \in \mathcal{S}} (\beta_i^t - \alpha_i^t)h_i^t\|_2 \quad \triangleright \text{Equation 30, 34} \tag{54}$$

$$\leq \sqrt{(1 - \eta\mu + 2\eta\gamma^t + \eta^2 L^2 + 2\eta^2 L\gamma^t)\|\Theta^t - \Theta_c^t\|_2^2 + 2\eta\gamma^t(1 + \eta L + 2\eta\gamma^t)} \tag{55}$$

$$+ 2\eta r^t M \quad \triangleright \text{Equation 49, 52, 53} \tag{56}$$

$$\leq \sqrt{1 - \eta\mu + 2\eta\gamma^t + \eta^2 L^2 + 2\eta^2 L\gamma^t}\|\Theta^t - \Theta_c^t\|_2 + \sqrt{2\eta\gamma^t(1 + \eta L + 2\eta\gamma^t)} + 2\eta r^t M, \tag{57}$$

where the last inequality holds due to the fact that $\sqrt{a + b} \leq \sqrt{a} + \sqrt{b}$ for $\forall a \geq 0$ and $\forall b \geq 0$, which completes our proof for Lemma 1.

## A.2 Proof of Theorem 1

We denote $A_t = \sqrt{1 - \eta\mu + 2\eta\gamma^t + \eta^2 L^2 + 2\eta^2 L\gamma^t}$, $A = \sqrt{1 - \eta\mu + 2\eta\gamma + \eta^2 L^2 + 2\eta^2 L\gamma}$, $B_t = \sqrt{2\eta\gamma^t(1 + \eta L + 2\eta\gamma^t)} + 2\eta r^t M$, and $B = \sqrt{2\eta\gamma(1 + \eta L + 2\eta\gamma)} + 2\eta r M$. Since $\gamma^t \leq \gamma$ and $r^t \leq r$, we have $A_t \leq A$ and $B_t \leq B$. Thus, based on Lemma 1, we have:

$$\|\Theta^t - \Theta_c^t\|_2 \leq A\|\Theta^{t-1} - \Theta_c^{t-1}\|_2 + B. \tag{58}$$

Then, we can iteratively apply the above equation to prove our theorem. In particular, we have:

$$\|\Theta^t - \Theta_c^t\|_2$$

$$\leq A\|\Theta^{t-1} - \Theta_c^{t-1}\|_2 + B \tag{59}$$

$$\leq A(A\|\Theta^{t-2} - \Theta_c^{t-2}\|_2 + B) + B \tag{60}$$

$$= A^2\|\Theta^{t-2} - \Theta_c^{t-2}\|_2 + (A^1 + A^0)B \tag{61}$$

$$\leq A^t\|\Theta^0 - \Theta_c^0\|_2 + (A^{t-1} + A^{t-2} + \cdots + A^0)B \tag{62}$$

$$= A^t\|\Theta^0 - \Theta_c^0\|_2 + \frac{1 - A^t}{1 - A}B \tag{63}$$

$$= (\sqrt{1 - \eta\mu + 2\eta\gamma + \eta^2 L^2 + 2\eta^2 L\gamma})^t\|\Theta^0 - \Theta_c^0\|_2$$

$$+ \frac{1 - (\sqrt{1 - \eta\mu + 2\eta\gamma + \eta^2 L^2 + 2\eta^2 L\gamma})^t}{1 - \sqrt{1 - \eta\mu + 2\eta\gamma + \eta^2 L^2 + 2\eta^2 L\gamma}}(\sqrt{2\eta\gamma(1 + \eta L + 2\eta\gamma)} + 2\eta r M), \tag{64}$$

When the learning rate satisfies $0 < \eta < \frac{\mu - 2\gamma}{L^2 + 2L\gamma}$, we have that $0 < 1 - \eta\mu + 2\eta\gamma + \eta^2 L^2 + 2\eta^2 L\gamma < 1$. Therefore, the upper bound becomes $\frac{\sqrt{2\eta\gamma(1 + \eta L + 2\eta\gamma)} + 2\eta r M}{1 - \sqrt{1 - \eta\mu + 2\eta\gamma + \eta^2 L^2 + 2\eta^2 L\gamma}}$ as $t \to \infty$. Hence, we prove our Theorem 1.

# B Complete Algorithms

## B.1 Complete Algorithm of FLGAME

Algorithm 1 shows the complete algorithm of FLGAME. In Line 1, we construct an auxiliary global model. In Line 2, the function REVERSEENGINEER is used to reverse engineer the backdoor trigger and target class. In Line 4, we compute the local model of client $i$ based on its local model update. In Line 5, we compute a genuine score for client $i$. In Line 6, we update the global model based on genuine scores and local model updates of clients.

## B.2 Complete Algorithm for a Compromised Client

Algorithm 2 shows the complete algorithm for a compromised client. In Line 1, we randomly subsample $\rho_i$ fraction of training data from $\mathcal{D}_i$. In Line 5, the function CREATEBACKDOOREDDATA

---

**Algorithm 1:** FLGAME

---

**Input:** $\Theta^t$ (global model in the $t$th communication round), $g_i^t, i \in \mathcal{S}$ (local model updates of
      clients), $\mathcal{D}_s$ (clean training dataset of server), $\eta$ (learning rate of global model).

**Output:** $\Theta^{t+1}$ (global model for the $(t+1)$th communication round)

1   $\Theta_a^t = \Theta^t + \frac{1}{|\mathcal{S}|}\sum_{i \in \mathcal{S}} g_i^t$

2   $\delta_{re}, y_{re}^{tc} = \text{REVERSEENGINEER}(\Theta_a^t)$

3   **for** $i \in \mathcal{S}$ **do**

4      $\Theta_i^t = \Theta^t + g_i^t$

5      $p_i^t = 1 - \frac{1}{|\mathcal{D}_s|}\sum_{\mathbf{x} \in \mathcal{D}_s} \mathbb{I}(G(\mathbf{x} \oplus \delta_{re}; \Theta_i^t) = y_{re}^{tc})$

6   $\Theta^{t+1} = \Theta^t + \eta \frac{1}{\sum_{i \in \mathcal{S}} p_i^t} \sum_{i \in \mathcal{S}} p_i^t g_i^t$

7   **return** $\Theta^{t+1}$

---

is used to generate backdoored training examples by embedding the backdoor trigger $\delta$ to $\lfloor \min(j * \zeta, 1)|\mathcal{D}_i \setminus \mathcal{D}_i^{rev}|\rfloor$ training examples in $\mathcal{D}_i \setminus \mathcal{D}_i^{rev}$ and relabel them as $y^{tc}$, where $|\cdot|$ measures the number of elements in a set. In Line 6, the function TRAININGLOCALMODEL is used to train the local model on the training dataset $\mathcal{D}_i' \cup \mathcal{D}_i \setminus \mathcal{D}_i^{rev}$. In Line 7, we estimate a genuine score. In Line 11, we use the function CREATEBACKDOOREDDATA to generate backdoored training examples by embedding the backdoor trigger $\delta$ to $\lfloor \min(o * \zeta, 1)|\mathcal{D}_i|\rfloor$ training examples in $\mathcal{D}_i$ and relabel them as $y^{tc}$. In Line 12, we use the function TRAININGLOCALMODEL to train a local model on the training dataset $\mathcal{D}_i' \cup \mathcal{D}_i$.

---

**Algorithm 2:** ALGORITHM FOR A COMPROMISED CLIENT

---

**Input:** $\Theta^t$ (global model in the $t$th communication round), $\mathcal{D}_i$ (local training dataset of client $i$),
      $\rho_i$ (fraction of reserved data to find optimal $r_i^t$), $\zeta$ (granularity of searching for $r_i^t$), $\delta$
      (backdoor trigger), $y^{tc}$ (target class), and $\lambda$ (hyperparameter).

**Output:** $g_i^t$ (local model update)

1   $\mathcal{D}_i^{rev} = \text{RANDOMSAMPLING}(\mathcal{D}_i, \rho_i)$

2   $count = \lceil \frac{1}{\zeta} \rceil$

3   $max\_value, o \leftarrow 0, 0$

4   **for** $j \leftarrow 0$ **to** $count$ **do**

5      $\mathcal{D}_i' = \text{CREATEBACKDOOREDDATA}(\mathcal{D}_i \setminus \mathcal{D}_i^{rev}, \delta, y^{tc}, \min(j * \zeta, 1))$

6      $\Theta_{ij} = \text{TRAININGLOCALMODEL}(\Theta^t, \mathcal{D}_i' \cup \mathcal{D}_i \setminus \mathcal{D}_i^{rev})$

7      $p_{ij} = 1 - \frac{1}{|\mathcal{D}_i^{rev}|}\sum_{\mathbf{x} \in \mathcal{D}_i^{rev}} \mathbb{I}(G(\mathbf{x} \oplus \delta; \Theta_{ij}) = y^{tc})$

8      **if** $p_{ij} + \lambda \min(j * \zeta, 1) > max\_value$ **then**

9         $o = j$

10       $max\_value = p_{ij} + \lambda \min(j * \zeta, 1)$

11   $\mathcal{D}_i' = \text{CREATEBACKDOOREDDATA}(\mathcal{D}_i, \delta, y^{tc}, \min(o * \zeta, 1))$

12   $\Theta_i^t = \text{TRAININGLOCALMODEL}(\Theta^t, \mathcal{D}_i' \cup \mathcal{D}_i)$

13   **return** $\Theta_i^t - \Theta^t$

---

## C   ADDITIONAL EXPERIMENTAL SETUP AND RESULTS

### C.1   ARCHITECTURE OF GLOBAL MODEL

Table 5 shows the global model architecture on MNIST dataset.

### C.2   VISUALIZATION OF GENUINE SCORE OF FLGAME AND TRUST SCORE OF FLTRUST (CAO ET AL., 2021A)

Our FLGAME computes a genuine score for each client which quantifies the extent to which a client is benign in each communication round. Intuitively, our FLGAME would be effective if the genuine score is small for a compromised client but is large for a benign one. FLTrust (Cao et al., 2021a)

**Table 5: Architecture of the convolutional neural network for MNIST.**

| Type | Parameters |
|---|---|
| Convolution | $3 \times 3$, stride=1, 16 kernels |
| Activation | ReLU |
| Max Pooling | $2 \times 2$ |
| Convolution | $4 \times 4$, stride=2, 32 kernels |
| Activation | ReLU |
| Max Pooling | $2 \times 2$ |
| Fully Connected | $800 \times 500$ |
| Activation | ReLU |
| Fully Connected | $500 \times 10$ |

computes a trust score for each client in each communication round. Similarly, FLTrust would be effective if the trust score is small for a compromised client but is large for a benign one. Figure 3 visualizes the average genuine or trust scores for compromised and benign clients of FLGAME and FLTrust on MNIST dataset. We have the following observations from the figures. First, the average genuine score computed by FLGAME drops to 0 quickly for compromised clients. In contrast, the average trust score computed by FLTrust drops slowly. Second, the average genuine score computed by FLGAME for benign clients first increases and then becomes stable. In contrast, the average genuine score computed by FLTrust for benign clients decreases as the number of iterations increases. As a result, our FLGAME outperforms FLTrust.

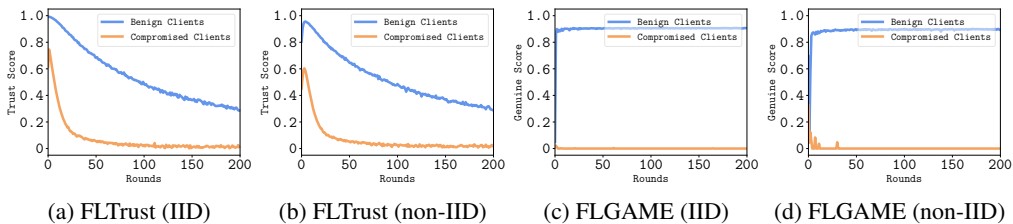

(a) FLTrust (IID)     (b) FLTrust (non-IID)     (c) FLGAME (IID)     (d) FLGAME (non-IID)

**Figure 3: (a)(b): Average trust scores computed by the server for benign and compromised clients of FLTrust on MNIST in IID and non-IID settings under Scaling attack. (c)(d): Average genuine scores computed by the server for benign and compromised clients of FLGAME on MNIST in IID and non-IID settings under Scaling attack. The clean training dataset of the server is the same for FLTrust and FLGAME.**

