# OpenReview forum: "FLGAME: A Game-theoretic Defense against Backdoor Attacks In Federated Learning"
_ICLR.cc/2023/Conference — Submitted to ICLR 2023_

### Official Review · Reviewer_tQyp · 2022-10-24

**Confidence:** 3
**Correctness:** 3
**Technical Novelty And Significance:** 3
**Empirical Novelty And Significance:** 2
**Recommendation:** 5

**Clarity, Quality, Novelty And Reproducibility:**

The discussion of the attack/defense used was clear. The paper and experiments are reproducible.

**Strength And Weaknesses:**

Strengths:
•	FLGAME achieves relatively high model accuracy with a very low attack success rate compared to prior defense.
•	The dataset the server has does not need to be in-domain. This is a much easier requirement to achieve that prior work.
•	The introduction of using an adaptive defense is a good contribution.

Weaknesses:
•	The experimental datasets were very simple. I would like to see more complex datasets, such as ImageNet/Tiny Imagenet.
•	Please expand on the contribution of the compromised clients in the model update. It’s not clear whether the attack success rate is low because the compromised clients have a low genuine score or if their updates result in weak backdoor success.
•	It is okay to include a few baseline comparisons. However, many of the defenses compared were not intended for backdoor attacks. Please show comparisons to other new defenses aimed for backdoor defense.
•	The paper relies on prior work for reverse engineering backdoor triggers and target class. I would like to see more about this. What limitations does this have? If the compromised clients use larger L1 norm triggers, does this fail?


**Summary Of The Paper:**

This paper introduces FLGAME, a defense against adaptive backdoor attacks. It formulates the compromised client attack and server defense as a form of minimax optimization where the clients want to maximize their contribution to the model update and the server wants to minimize malicious contribution. This optimization centers around a genuine score that weights the model update during aggregation. The server computes this through a reverse engineered backdoor trigger and target class.

**Summary Of The Review:**

The method of using a minimax game and the use of an adaptive defense is a very interesting and important contribution. However, my main issue is the lack of depth in the empirical results. In particular, the datasets used and the analysis on compromised client participation can be made much stronger. Please refer to the weaknesses section for specifics.

---

> ### Author Response · Authors · 2022-11-18
> **Official Response to Reviewer tQyp**
>
> Thanks a lot for the constructive feedback!
>
> Comment 1: The experimental datasets are very simple.
> Response 1: We note that we adopt benchmark datasets that are widely used to study backdoor attacks and their defenses in federated learning. We will add more discussion in our revision.
>
> Comment 2: The reason why ASR is low is not clear.
> Response 2: Sorry for the confusion. The ASR is low because the genuine scores for compromised clients are small. Note that the impact of the local model of a client on the global model is small if the genuine score for the client is small. Figure 3 in the Appendix visualizes the genuine scores for benign and compromised clients in each communication round. As the results show, the genuine scores for compromised clients are very low.
>
> Comment 3: Many compared baselines are not for backdoor attacks. Compared with new defenses that are designed for backdoor attacks.
> Response 3: We note that Norm-Clipping [2], DP [2], and FLTrust [3] are designed for backdoor attacks. Moreover,  FLTrust is $\textit{state-of-the-art}$ defense against backdoor attacks on federated learning. We follow the suggestion to compare with another SOTA baseline DeepSight [1]. Under the same setting, the ASR of DeepSight and our FLGAME are 12.61% and 10.73%, respectively. Our results indicate that FLGAME outperforms DeepSight.
>
> Comment 4: Will the defense fail for a trigger with a large L1 norm?
> Response 4: We have added the experiments as suggested. For block triggers of size $4\times 4$, $6\times 6$ and $8\times 8$, the attack success rates of our method are 10.24%, 10.98%, 10.31% and 13.69%, 13.23%, 15.80% for FLTrust respectively. We note that the backdoor attack is less stealthy when an attacker adopts a larger trigger.
>
> [1] Rieger et al., "DeepSight: Mitigating Backdoor Attacks in Federated Learning Through Deep Model Inspection." NDSS. 2022.
> [2] Sun et al. "Can You Really Backdoor Federated Learning?" NeurIPS workshop. 2019.
> [3] Cao et al. “FLTrust: Byzantine-robust Federated Learning via Trust Bootstrapping.” NDSS. 2021.

---

### Official Review · Reviewer_Z2ep · 2022-10-24

**Confidence:** 4
**Correctness:** 2
**Technical Novelty And Significance:** 1
**Empirical Novelty And Significance:** 1
**Recommendation:** 1

**Clarity, Quality, Novelty And Reproducibility:**


The paper is easy to follow; the proposed method is seriously flawed in my view;

**Strength And Weaknesses:**


Sorry that I don't see any strength in the paper.

The proposed method is seriously flawed in my view for following reasons:

- the assumption that the server is able to access and inspect local model updates poses privacy leakage risks on private data in the face of deep leakage type of gradient attacks [1];  this serious privacy risk defeats the purpose of federated learning in the first place;  authors are advised to reconsider the threat model and take into account various privacy-preserving mechanisms such as Differential privacy or Homomorphic Encryption;

[1] Deep Leakage from Gradients, NeurIPS 2019;

- the proposed method evaluates local model updates based on the agreement with the reverse-engineered trigger sets and target classes; nevertheless, the approach does not make sense at all since the target class selected by the reverse-engineering process is by no means the same as the target class might be selected by real backdoor attackers; the paper proposes to “find the backdoor trigger and target class such that the genuine scores for benign clients are large but they are small for compromised clients”， however, the compromised clients are a prior unknown, and the proposed method cannot reliably estimate the target class and solve this chicken and egg problem;

- the proposed method is inefficient in the evaluation of local model update (Line 4-5 in Algo. 1); this method cannot be efficiently applied to a federated learning scenario with large number of clients; that is probably why experiments reported in the paper only used 10 and 20 clients, which are far less than typical cross-devices use cases with millions of clients;

**Summary Of The Paper:**

This paper proposes to formulate the defense against backdoor attacks as a minimax game between the server (defender) and attackers in federated learning. The minmax problem is solved in three steps by 1) construct an auxilary global model; b) reverse-engineer backdoor samples with selected target classes; c) assign to each client a genuine score is used to weigh model update uploaded by the client.


**Summary Of The Review:**

The proposed method is seriously flawed in my view for following reasons:

- the assumption that the server is able to access and inspect local model udpates imposes privacy risks on private data in the face of deep leakage type of gradient attacks [1];  this serious privacy risk defeats the purpose of federated learning in the first place;  authors are advised to reconsider the threat model and take into account various privacy-preserving mechanisms such as Differential privacy or Homomorphic Encryption;

[1] Deep Leakage from Gradients, NeurIPS 2019;

- the proposed method evluates local model udpates based on the agreement with the reverse-engineered trigger sets and target classes; nevertheless, the approach does not make sense at all since the target class selected by the reverse-engineering process is by no means the same as the target class might be selected by real backdoor attackers; the paper proposes to “find the backdoor trigger and target class such that the genuine scores for benign clients are large but they are small for compromised clients”， however, the compromised clients are a prior unknown, and the proposed method cannot reliably estimate the target class and solve this checken and egg problem;

- the proposed method is inefficient in the evaluation of local model udpate (Line 4-5 in Algo. 1); this method cannot be efficiently applied to a federated learning scenario with large number of clients; that is probably why experiments reported in the paper only used 10 and 20 clients, which are far less than typcial cross-devices use cases with millions of clients;

---

> ### Author Response · Authors · 2022-11-18
> **Official Response to Reviewer Z2ep**
>
> Thanks a lot for the constructive feedback!
>
> Comment 1: The threat model is strong, e.g., the access to local model updates poses privacy leakage.
> Response 1: First of all, we do not assume the server directly inspects the gradients from the clients (FedSGD). Instead, we consider FedAvg, where the local model update $\theta’-\theta$ is sent to the server. Such a setting is standard in federated learning [1].  Besides, our threat model is consistent with the previous work on backdoor attacks and their defenses [2,3]. We note that differential privacy may hurt the utility of global models, and homomorphic encryption may incur large computation and communication costs. We will add the discussion to our paper.
>
> Comment 2: The compromised clients are unknown.
> Response 2: Sorry for the confusion. Our defense does not require the defender to know the compromised clients. When we reverse engineer the backdoor trigger and target class, we construct an auxiliary global model which aggregates local model updates from all clients. Then, we reverse engineer the backdoor trigger and target class based on it. Our experimental results show that FLGAME can reliably estimate them. For instance, the genuine scores for malicious clients are very low (close to 0), as visualized in Figure 3 in Appendix. We will clarify.
>
> Comment 3: The proposed method is inefficient.
> Response 3: On a single NVIDIA 2080 Ti GPU, it takes 0.56 seconds to evaluate one local model. We note that the server is usually a resourceful tech company (e.g., Google, Meta, Apple), which would have enough computation resources to evaluate local models on millions of clients. Moreover, those local models can be evaluated in parallel. We will discuss it.
>
> [1] McMahan et al. “Communication-efficient learning of deep networks from decentralized data.” Artificial Intelligence and Statistics. 2017.
> [2]  Bagdasaryan et al. “How to backdoor federated learning.” International Conference on Artificial Intelligence and Statistics. 2020.
> [3] Cao et al. “FLtrust: Byzantine-robust federated learning via trust bootstrapping.” NDSS. 2021.

---

### Official Review · Reviewer_TbmV · 2022-10-25

**Confidence:** 4
**Correctness:** 2
**Technical Novelty And Significance:** 2
**Empirical Novelty And Significance:** 3
**Recommendation:** 5

**Clarity, Quality, Novelty And Reproducibility:**

I think the analysis of this paper is not very general. The paper restricts
attackers and their capabilities to its designed minimax game model. This
paper proposes a minimax game model between defenders and attackers, where
both optimize a genuine score of backdoored and clean models. However, in a
practical scenario, the attack methods are unavailable, so attackers don't
need to optimize the genuine scores. So this game model considers a restricted
assumption of attackers' attack methods, making its comparison with other
baselines less convincing.

The theoretical analysis to guarantee the robustness of its method is based on
an impractical assumption. In backdoor attacks, the common triggers are not
bounded by $L_p$ distances. This paper assumes that the Lipschitz continuous
gradient will bound the backdoor updates of the loss function. It requires the
backdoor attacks in the FL model to be as strong as possible to make sure it
can be bounded. Existing works consider a successful backdoor attack (e.g.,
attack success rate over 90%) rather than consider the strongest backdoor
attack. Stronger attacks are more likely to be detected. So I think this
assumption is not practical.

The proposed defense method has limited technical novelty as it mostly applies
a typical reverse-engineering method (Neural Cleanse). There are many existing
backdoor attacks like WaNet [3], the invisible attack [4], which can bypass
Neural Cleanse. These attacks consider trigger patterns other than patches to
break the assumptions of Neural Cleanse. So it is also important to consider
these attacks in this paper, which can help enhance the effectiveness of its
method.

[1] Gu et al. "BadNets: Identifying Vulnerabilities in the Machine Learning Model Supply Chain" IEEE Access 2019
[2] Liu et al. "Trojaning Attack on Neural Networks" NDSS 2018
[3] Nguyen et al. "WaNet - Imperceptible Warping-based Backdoor Attack" ICLR 2021
[4] Li et al. "Invisible Backdoor Attack with Sample-Specific Triggers" ICCV 2021


**Strength And Weaknesses:**

It is an interesting angle to look at the federated learning backdoor problem.
The analysis is well presented.

Its discussion is limited to strong assumptions about the adversary.
The theoretical analysis makes impractical assumptions about backdoor attacks.
The proposed defense has limited novelty.

**Summary Of The Paper:**

This paper proposes a minimax game model of backdoor attacks between attackers
and defenders in the federated learning (FL) paradigm.  Based on the analysis
of this model, it uses a reverse-engineering technique to defend against
backdoor attacks in the FL process. It provides theoretical analysis and
experimental results to present the effectiveness of its defense mechanism.


**Summary Of The Review:**

This paper makes strong assumptions about attackers, and it lacks technical novelty (proposed defense).

---

> ### Author Response · Authors · 2022-11-18
> **Official Response to Reviewer TbmV**
>
>
> Thanks a lot for the constructive feedback!
>
> Comment 1: Restricted attacks.
> Response 1: Sorry for the confusion. From the defender's perspective, the defender doesn't have information about attacks at all, just as other baselines. Note that our minimax game is for analysis purpose. When we solve the optimization problem for the defender, we do not assume any information about attacks.  We also follow the suggestion to evaluate our defense under existing backdoor attacks where the attacker does not optimize the genuine scores. The attack success rate is 9.75%, which means our defense is consistently effective under static attacks.
>
> Comment 2: Theoretical analysis.
> Response 2: Thanks for pointing it out. We note that those assumptions are also made by previous work on defending against backdoor attacks to federated learning [1, 2]. We add experiments in which we evaluate our defense under existing attacks (see Response 1 for results). Our experimental results show that our FLGAME can effectively defend against those attacks.
>
> Comment 3: Other backdoor attacks like WaNet and Invisible attacks.
> Response 3: Thanks for the references. We will discuss them as suggested. In our experiments, we use the triggers from previous work [3]. For instance, the trigger is a ‘X’ pattern for MNIST, which is not a patch. Our results demonstrate our defense is effective with non-patch patterns. We will clarify.
>
> [1] Xie et al. “CRFL: Certifiably Robust Federated Learning against Backdoor Attacks.” ICML. 2021.
> [2] Cao et al. “FLTrust: Byzantine-robust Federated Learning via Trust Bootstrapping.” NDSS. 2021.
> [3] Bagdasaryan et al. "How to backdoor federated learning." International Conference on Artificial Intelligence and Statistics. 2020.

---

### Official Review · Reviewer_jdkR · 2022-10-25

**Confidence:** 4
**Correctness:** 4
**Technical Novelty And Significance:** 2
**Empirical Novelty And Significance:** 2
**Recommendation:** 5

**Clarity, Quality, Novelty And Reproducibility:**

The writing quality is adequate and the paper is mostly easy to follow. However, it is worth mentioning in the paper, whether the backdoor attacks corresponding to different defenses are the same or not. For the baselines, I believe the same backdoor attacks can be leveraged. But for the proposed defense, since it involves dynamic game between the defender and attacker, the attacks are actually changing.

As for the originality, this work is not the first to propose defenses based on reputation scores of the clients. In addition, the reputation score computation itself vulnerable to adaptive attacks, and I am concerned that the contribution is limited.

**Strength And Weaknesses:**

Strength:
The paper is well written and the idea of defending against adaptive attackers is impressive.
Weakness:
1. the whole framework relies on the defender is able to reverse engineer useful backdoor triggers so as to effectively minimize the impact from compromised clients. However, this design may be exploited by the attacker [1]. Specifically, it is possible that attacker generate backdoors in a way that, either the reversed engineered backdoor samples are not effective or are similarly effective for both the compromised and benign clients. This assumption is the major weakness of this paper.
2. it is unclear how well those theories (e.g., Theorem 1) imply in practice. For example, the assumption on $r_i^t=0$ seems to not hold in practice and so, the backdoor model cannot be the same the clean global model, even asymptotically as the training rounds continues.
3. the threat model makes too many restrictions on the attacker capability. For example, the attackers can only augment the training data with backdoored samples while in practice, they may choose to replace some samples with backdoored samples. The attackers are also assumed to be unaware of the benign scores of other clients while in practice, they may be able to simulate the benign scores (given the knowledge that the server is calculating benign scores for each client using reversed engineered triggers). In the worse case scenario, powerful attackers may also be able to get the actual benign scores. It is also assumed that attackers can only use its own backdoor trigger when deciding the proper poisoning ratio while in practice, attackers can also simulate the reverse engineering process of the defender.
4. there are also some unrealistic experimental settings in the paper. For example, it is assumed that all clients in each round will be selected while in practice, defenders usually randomly selects subset of the clients. Also, the fraction of compromised clients are mostly evaluated under 60%, which, theoretically, should be hopeless to defend against. The lowest fraction of compromised clients is still 20%, which is quite a high number. I am curious to see the comparison among different defenses (not just FLTrust and the proposed method) under even lower compromised clients.
5. some recent works are missing from the paper. For example, the most recent defense on backdoor attacks in federated learning [2].
[1] Veldanda et al., "On Evaluating Neural Network Backdoor Defenses".
[2] Rieger et al., "DeepSight: Mitigating Backdoor Attacks in Federated Learning Through Deep Model Inspection", NDSS 2022.

**Summary Of The Paper:**

This paper proposes a dynamic backdoor defenses in federated learning, where both the defender and the attacker can dynamically adjust their strategies as the federated training continues. The defender reverse engineers the potential backdoor trigger so as to finally generate the benign score for each client, and downweights the clients with lower benign scores. Attackers aim to find the proper backdoor poisoning ratio such that a good tradeoff between backdoor effectiveness and benign scores of compromised clients can be reached.

**Summary Of The Review:**

The idea of defending against adaptive backdoor attacks in federated learning is interesting. However, the proposed framework did not consider this "adaptive" aspect thoroughly and hence, is not convincing. Also, the experimental settings are not practical and some recent baselines are missing. Considering all the weakness mentioned above, rejection is given.

---

> ### Author Response · Authors · 2022-11-18
> **Official Response to Reviewer jdkR**
>
> Thanks a lot for the constructive feedback!
>
> Comment 1: The proposed defense can be exploited by the attacker.
> Response 1: Thanks for the suggestion. Indeed, the attacker can be adaptive, and therefore we consider the game theoretic analysis in this work, allowing the attacker to play the best responses in terms of the number of poisoning instances. It would be interesting to consider that the attacker can optimize its trigger patterns as well, but since existing works [1] show that any trigger (e.g., even a simple four-pixel trigger) is effective to backdoor different ML models, we believe that optimizing the number of poisoning instances would help to construct a more sophisticated attacker. This way, we can see that our defense method is effective against a stronger attacker.
> In addition, we indeed evaluate different triggers following the suggestions, and we do observe that our method consistently outperforms the $\textit{state-of-the-art}$ baseline FLTrust. For block triggers of size $4\times 4$, $6\times 6$, and $8\times 8$, the attack success rates of FLGAME and FLTrust are (10.24%, 13.69%), (10.98%, 13.23%), (10.31%, 15.80%), i.e., our FLGAME consistently outperforms FLTrust for different triggers. We will make such discussions clear and add our results in our revision.
>
> Comment 2: Whether the assumptions made on Theorem 1 are practical, e.g.,$r_i^t=0$.
> Response 2: In our Theorem 1, we don’t assume $r_i^t=0$. Our Theorem 1 implies that the global model parameters under our defense do not deviate too much from those of the global model without attack when $r_i^t$ is bounded. Moreover, when $r_i^t \neq 0$, the local training data of malicious clients would change. As a result, the global model parameters would be different from those of the clean global model in which $r_i^t=0$ for each malicious client. We will clarify it.
>
> Comment 3: The attacker can replace training samples with backdoored samples.
> Response 3: Thanks for the comment. We consider that the $r_i^t$ fraction of training samples of malicious clients are replaced by backdoored samples and optimize it under our game-theoretic framework. Under our default setting, the attack success rate of FLGAME is 9.71%. The results demonstrate that our defense is also effective. We will add the results to our paper.
>
> Comment 4: Unrealistic experimental settings, e.g., all clients are selected in each round; comparison under a low fraction of malicious clients.
> Response 4: We added experiments where only a subset of clients are selected. In particular, we consider that 10 clients are selected in each communication round while the total number of participants is 100. We find that our method can achieve a 10.01% attack success rate under the default setting, which means our defense is still effective when a small fraction of clients is selected in each communication round. We find that our defense is comparable with robust aggregation methods and FLTrust when the fraction of malicious clients is low. However, when the fraction of malicious clients is large, our defense is still effective, while existing defenses are not.
>
> Comment 5: Missing references.
> Response 5: Thanks for the references. We will add them. Moreover, we follow the suggestion to compare with the recent SOTA baseline DeepSight [2]. Under the same setting, the ASRs of DeepSight and our FLGAME are 12.61% and 10.73%, respectively. Our results indicate that FLGAME outperforms DeepSight.
>
> Comment 6: Whether attacks corresponding to different defenses are the same.
> Response 6: Sorry for the confusion. We consider the same attack, which does not change, for FedAvg, Krum, Median, Norm-Clipping, DP, and FLTrust. For our FLGAME, we consider dynamic attacks. We also evaluate our FLGAME when the attack does not change. The attack success rate is 9.75% under our defense in the default setting. Our results show that our FLGAME is still effective under static attack. We will clarify as suggested.
>
> [1] Xie et al. "Dba: Distributed backdoor attacks against federated learning." ICLR. 2019.
> [2] Rieger et al. "DeepSight: Mitigating Backdoor Attacks in Federated Learning Through Deep Model Inspection." NDSS. 2022.

---

> > ### Comment · Reviewer_jdkR · 2022-12-05
> > **Thanks for the clarification**
> >
> > Thanks the authors clarifying most of my concerns and also performing new experiments. However, my major concern still remains on whether the proposed defense is free from future attacks. It is good formulate the problem from the perspective of game theory, but game theoretic formulation is way too general and of needs to define the action space concretely. My concern is more on, whether this action space completely covers the possible actions that can happen in practice. To be more specific, constraining attackers to be able to reverse engineer the backdoor trigger sounds like a strong assumption to me, because adaptive attackers can leverage this insight to design evasive attacks. Of course, the authors can argue that existing attacks do not work well against these, but I am more worried about some new attacks after this work is published. Given that some of my other concerns are addressed, I am raising the score.

---

### Author Response · Authors · 2022-11-18
**Revision Summary**

We thank the reviewers for the valuable suggestions and constructive feedback. We have added the following experiments in our rebuttal:
1. We evaluate our defense and compare it with the \textit{state-of-the-art} baseline FLTrust for different trigger sizes ($4\times 4$, $6\times 6$, and $8\times 8$). We find that our defense is consistently effective and better than FLTrust for different trigger sizes.
2. We evaluate our FLGAME when an attacker replaces training samples with backdoored samples. Our results show our defense is also effective.
3. We evaluate our FLGAME when only a small number of clients are selected in each communication round. Our results show FLGAME is effective in this setting.
4. We compare our FLGAME with a new $\textit{state-of-the-art}$ baseline DeepSight. We find that our defense outperforms it.
5. We evaluate our FLGAME under static attacks, i.e., we don’t leverage our game-theoretic framework to dynamically optimize attack strategy. Our results show our defense is effective against these attacks.
6. We evaluate the running time of evaluating local models of clients for the server.

We have also made several clarifications about our method:
1. We explain why we consider optimizing the number of poisoning instances.
2. We clarify and explain the assumption made on our Theorem 1.
3. We explain the results when the fraction of malicious clients is low.
4. We clarify the information required for the defender of our defense, e.g., we don’t require information about the attack method and compromised clients.
5. We clarify our threat model.
6. We clarify the attacks that we consider for evaluating our framework.
7. We explain why the ASR of backdoor attacks under our defense is low.
8. We clarify that a non-patch trigger is also used in our evaluation.

---

### Decision · Program_Chairs · 2023-01-20

**Decision:**

Reject

**Justification For Why Not Higher Score:**

The reviewers identified some weaknesses that the authors attempted to address in their rebuttal, but the major concern remains whether the proposed defense is general, and strong enough.

**Justification For Why Not Lower Score:**

N/A

**Metareview: Summary, Strengths And Weaknesses:**

In this work, a minimax game is used to design an interactive defense mechanism called FLGAME which is proven to be theoretically robust and empirically more effective than state-of-the-art baselines. FLGAME can effectively defend against strategic attackers.

the reviewers identified the following strengths and weaknesses

Strengths:
• FLGAME achieves relatively good accuracy with a low attack success rate compared to prior defense.
• The dataset the server has does not need to be in-domain.
• The use of an adaptive defense mechanism.

Weaknesses:
• The experimental datasets are very simple.
• The set of baselines tested against are questionable.
• The paper relies heavily for reverse engineering backdoor triggers and target class. This may lead to a weak defense mechanism
• The proposed method evaluates local model updates based on the agreement with the reverse-engineered trigger sets and target classes.  this does not accurately simulate a real-world backdoor attack.
• The proposed method can be inefficient when evaluating model updates.
• The assumption that the server is able to access/inspects local updates poses privacy risks.

In the rebuttal, the authors evaluated their defense against different trigger sizes, evaluated it when an attacker replaces training samples with backdoored samples, compared it with the state-of-the-art baseline DeepSight, evaluated it under static attacks, and evaluated the running time of evaluating local models of clients for the server. Additionally, the authors clarified and explained assumptions made, results when the fraction of malicious clients is low, the information required for the defender of their defense, their threat model, and the attacks they consider for evaluating their framework.

However, the reviewers' major concern still remained whether the proposed defense could withstand future attacks. One suggested that the authors should concretely define the action space to cover possible actions, and consider the possibility of new attacks once the work is published.

I also stand by the comments made from the reviewers, and believe this paper is not ready for publication, but does have the potential, if the above concerns are thoroughly addressed.

**Summary Of Ac-Reviewer Meeting:**

N/A